# Comparing Community Needs and REDD+ Activities for Capacity Building and Forest Protection in the Équateur Province of the Democratic Republic of Congo

Edward A. Morgan [1,2,*], Glenn Bush [3], Joseph Zambo Mandea [4], Melaine Kermarc [3] and Brendan Mackey [1,5]

1 Griffith Climate Change Response Program, Griffith University, Southport, QLD 4222, Australia; b.mackey@griffith.edu.au
2 Cities Research Institute, Griffith University, Nathan, QLD 4222, Australia
3 Woodwell Climate Research Center, Woods Hole, Falmouth, MA 02540-1644, USA; gbush@woodwellclimate.org (G.B.); kermarcm@gmail.com (M.K.)
4 Woodwell Climate Research Center, Mbandaka, Democratic Republic of the Congo; jzambo@woodwellclimate.org
5 Climate Action Beacon, Griffith University, Southport, QLD 4222, Australia
* Correspondence: ed.morgan@griffith.edu.au; Tel.: +61-(0)7-3735-9248

**Abstract:** Primary forests are essential ecosystems that can play a key role in mitigating climate change. REDD+ is designed to help countries and communities secure benefits for avoiding deforestation but has faced significant implementation challenges. There are substantial potential benefits for REDD+ in the Democratic Republic of Congo (DRC), where shifting agriculture is the major cause of deforestation. However, implementation requires significant capacity building in a number of sectors and at a number of levels. This paper explores how well the capacity building activities within the DRC REDD+ strategy are aligned with the capacity needs identified by provincial government stakeholders and local communities in the Équateur province of the DRC, identified through workshops and surveys. The research suggests that while many technical capacity needs identified by stakeholders could be potentially addressed by the REDD+ strategy, there are number of systemic capacity needs that are unlikely to be addressed. Failure to address these needs risks undermining any implementation of REDD+. The results suggest that education and training in governance and management, as well as fundamental education in sustainability, are key capacity needs that REDD+ may need to incorporate. The results also provide further evidence that REDD+ projects need to be long-term and take into account the local context and needs in order to be effective.

**Keywords:** forest landscapes; REDD+; capacity building; community needs; Democratic Republic of the Congo

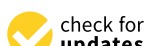



## 1. Introduction

Forests are vital ecosystems at a global, regional and local scale, and are especially important for responding to climate change and protecting biodiversity [1–3]. In particular, primary forests—those not subject to management for commodity production and other industrial scale commercial uses and whose structure and function are dominated by natural processes [4]—provide a greater array of high quality services, compared to secondary growth forests or plantations [5–7]. Consequently, protecting areas of primary forest needs to be a priority for forest management. The Democratic Republic of Congo (DRC) forest cover has over 100 million ha of primary forest, with 60% classified as within 'intact forest landscapes', and has the largest contiguous area of tropical forest outside of the Amazon [8].

However, primary forests are facing ongoing forest loss and degradation globally [9,10]. Commonly, this is due to extractive logging and mining or the clearing of forest for industrial agriculture [11,12]. In the Democratic Republic of Congo (DRC), on the other

hand, much of the deforestation is caused by shifting ('slash-and-burn') subsistence level agriculture [13,14].

Recognising the importance of forests for climate change mitigation generally, REDD+ is the UNFCCC process designed to limit emissions from deforestation and support reforestation for carbon storage [15,16]. However, implementing REDD+ has proven challenging [17–20]. One of the limitations of REDD+ has the been the need for significant capacity building to put in place the necessary technical knowledge and institutional governance arrangements to ensure monitoring, evaluation, verification, transparency and protection of human rights, and ultimately to ensure REDD+ results in reduced or avoided deforestation [15,16,21].

The objective of this paper is to compare the capacity building included within the DRC REDD+ strategy with the capacity needs identified by stakeholders at the local and provincial level. This paper explores the types of capacity needed at the provincial level in the Équateur province of the DRC. Using data from workshops with provincial government stakeholders, the study compares capacity building challenges and needs identified by stakeholders at the provincial level with the DRC REDD+ strategy and Investment Plan, to identify overlaps and to highlight gaps in the REDD+ strategy. The research will also help provincial government stakeholders identify opportunities to address their context-specific concerns through REDD+ capacity building. The paper seeks to answer the question: Is REDD+ capacity building in the DRC addressing the needs of the provincial government and communities in the Équateur province? The challenges the Équateur province is facing are similar not only across the DRC, but also in many highly forested, low deforestation developing countries, and the results and conclusion are likely applicable in many other similar contexts.

After a description of the method, the paper provides a brief discussion of capacity building to provide an analytical framework to compare the DRC National REDD+ strategy with the issues identified by the provincial government. It then presents the results of the analysis, and compares and contrasts the national and provincial issues. Finally, the paper discusses the implications of the overlaps and gaps between the national goals and provincial issues.

## 1.1. REDD+, Forest Protection and Capacity Building

The concept of REDD+ is simple: tropical forested countries receive payments for preventing or reducing deforestation and for reforestation based on how much carbon these activities avoid emitting. However, although REDD+ as an outcome (reduced emissions) is straightforward, the REDD+ framework (the activities) to achieve the outcome have proved complex and have changed over time to mean a range of different things [15,22]. Countries agreed to rules and procedures for a new UNFCCC mechanism that credits emissions reducing activities in December 2021 at COP26, which have the potential to incentivise REDD+ projects. However, the exact relationship between compliance-based and voluntary markets, and the status of pre-existing credits, remain unclear [23]. It remains to be seen how this agreement will affect REDD+ practice, especially around safeguards for communities and human rights.

Much of the focus of REDD+ has been on 'readiness'—preparing countries to be able to implement REDD+ programs. This has included a significant focus on capacity building. Broadly, capacity building can be defined as "the sum of efforts needed to develop, enhance and utilize the skills of people and institutions to follow a path of sustainable development" (United Nations Development Programme, 2001 cited in Downs 2003, p. 186). The UNFCCC recognises that "Establishing climate-friendly patterns of sustainable development depends on a broad range of approaches" [24], which requires significant capacity that not all countries have. The Convention, the Kyoto Protocol and the Paris Agreement all recognise and emphasise the importance of capacity building.

The focus on readiness and capacity building has led to criticism that REDD+ has been running for more than 10 years and spent millions of dollars without any significant

impact on actually avoiding or reducing emissions or improving incomes for communities [15,17,18,25]. Others have argued that it has had a range of positive impacts and created significant opportunities for novel approaches to forest protection [15,22,26–28]. However, with readiness funding reducing, it remains unclear if the substantial capacity that REDD+ readiness was designed to build has been successfully achieved or appropriately targeted.

Implementing REDD+ has increasingly been seen as a governance challenge [16,26,29]. In terms of capacity, a number of authors have criticised REDD+ as failing to benefit communities [17,25], with much of the funding going to governments for capacity building, and for not adequately protecting the rights of communities [30,31]. The UNFCCC REDD+ approach requires countries to follow FPIC principles and ensure participatory approaches, but a number of governance challenges remain [16,28].

Although there is general agreement that capacity building is essential for REDD+, there is less consensus on exactly what capacity building is needed. Capacity building for REDD+ can include a wide range of activities, including building technical skills around monitoring, reporting and verifying forest stocks and deforestation, building capacity of organisations to access REDD+ schemes; capacity to implement FPIC and ensure community participation; and institutional capacity to ensure transparency and equitable benefit sharing. The contexts both between and within countries are likely to be very different and need different types of capacity building [24,32].

### 1.2. Forest Protection and REDD+ in the DRC and the Équateur Province

The Congo Basin has the largest contiguous area of tropical forest outside of the Amazon [8]. The Democratic Republic of Congo (DRC) has significant areas of forest (70% of the Congo basin forest area, with the majority of DRCs forest cover considered primary forest (estimated at around 105 million Ha in 2000), with 60% classified as within 'intact forest landscapes' [8]. These forests provide essential ecosystem services for the whole region [33]. The DRC also has a historically relatively low deforestation level when compared to other tropical forested nations. However, given the large area of forest, the total area of deforestation is large in actual terms. Between 2001 to 2020, the Democratic Republic of the Congo lost 15.9 Mha of tree cover, equivalent to a 8.0% decrease in tree cover since 2000, and 9.71 Gt of $CO_2$e emissions [34].

Molinario et al. [35] provided conclusive evidence demonstrating that, from 2000–2015, subsistence agriculture was the overwhelmingly dominant driver for forest clearing in the DRC. This is achieved through both the expansion of settled areas at the forest frontier and isolated "pioneer" clearings within intact forest. Less than 1% of clearing was directly attributable to land uses such as mining, plantations, and logging, showing that the contemporary impact of commercial land use operations in the DRC is negligible. However, both artisanal and large-scale commercial operations for logging, mining, and plantations do have a wider, indirect influence on land use change that goes beyond the area directly implicated in their operations. Molinario et al. [35] estimated that 12% of forest loss in terms of frontier expansion and 9% of loss from pioneer clearing was found to be within 5 km of mines, logging, or plantations, and hence industrial land uses are an important factor to consider in land use planning and sustainability development.

The Équateur province (as defined under the 2014 decentralization framework) of the DRC is the 8th most forested province in the DRC. From 2001 to 2020, Équateur lost 602 kha of tree cover, equivalent to a 6.2% decrease in tree cover since 2000, and 386 Mt of $CO_2$e emissions [36] (GFW, 2021). In the Équateur province, as in the rest of the country, the main driver of deforestation is shifting agriculture, with limited industrial or artisanal logging less of a concern [13]. Deforestation is relatively low, but still a significant threat to the forest and has been increasing [36]. Further development will bring with it increased risk of deforestation due to growth in commercial and industrial-scale agriculture, extractive logging and mining, and growing urbanisation [37,38]. Improved capacity can help provincial governments and the local population make informed choices if or when extractive industries arrive and grow.

Despite political challenges and ongoing conflict in the east of the country, the DRC has been implementing a REDD+ process and has developed a REDD+ strategy and investment plan [39,40]. Several large REDD+ projects and jurisdictional programs have been implemented, with USD 264 million committed to the REDD+ process between 2009 and 2014 [41]. There was limited progress between 2013 and 2018, but more activity after a change of government and a period of relative stability. Recent research has suggested that REDD+ has the possibility to "largely mitigate future carbon emissions" while providing significant economic benefit [42]. The expectation of REDD+ to generate funding for forest conservation and management is enormous. [43], but there are significant challenges for REDD+ implementation in the country [19,21,29,44]. REDD+ implementation has resulted in hybrid governance arrangements [44,45]. The involvement of the private sector and multilateral organisations has reduced government legitimacy to act, but reinforces the perception of government legitimacy because government is perceived to be in control. The DRC REDD+ process also faces challenges given the importance of rigorous and transparent monitoring and evaluation, to ensure that REDD+ results in genuine reduced emissions [46]. Developing Forest Reference Emission Levels (FRELs) at the project and national level has been highly politicised [41,45], and the national FREL is still under review due to data and methodological concerns, despite being submitted in 2018 [41].

The larger structural problems within the DRC are well-recognised [47] and political instability has resulted in poor, unstable governance, and limited and contradictory policy generally and around forest management [19,41]. A full discussion of these issues is beyond the scope of this paper, but it is important to note that these issues directly limit effective forest management and need to be addressed if resource management is to be sustainable in the DRC. We do not suggest that these problems can be easily solved and we acknowledge that these structural problems exist, but we note that successful action is possible, despite these challenges [48]. In fact, it has been argued that the structures put in place through REDD+ or other payments for ecosystem services schemes may help countries address wider structural governance issues [49]. Here, we discuss the role of capacity building in this context, to investigate its importance for REDD+ and more generally. Significant effort has been put into building structures and processes. Ensuring that those involved have the necessary capacity will be key to both REDD+ action in the current context [21] and the wider issue of addressing these structural challenges. This paper contributes to this by improving our understanding of what type of capacity building is needed at the provincial and community level for REDD+.

Capacity building has been, and continues to be, a significant part of the DRCs' REDD+ strategy. There are significant challenges to implementing REDD+ in the DRC, and it is widely recognised that the process is likely to be slow and require significant capacity building [21,44,50]. For a country such as DRC, building the necessary capacity is a significant challenge given its institutional weaknesses [19,47] and low levels of education [51]. This paper adds to the discussion of the capacity building challenge by investigating the overlaps and gaps between provincial government, industry and community capacity needs, and the DRC REDD+ strategy capacity building goals. The paper reports results of local stakeholder workshops and surveys to identify key challenges and opportunities in capacity building in the Équateur province.

## 2. Materials and Methods

### 2.1. Data Collection

The capacity needs of provincial government, expert and business stakeholders were identified using a collaborative problem-tree analysis [52], during workshops run as part of the 'Projet Équateur' REDD+ preparedness project. The workshops took place in Mbandaka, Équateur province, DRC, in 2016 and 2019, with the later workshop being used to validate and update the original problem trees after a change in the Provincial Government. The updated problem trees are presented in the Supplementary Information (Figures S1–S4); the workshop focused on five sectors, but only the problems trees focused on three key

issues were used in this analysis: agriculture, as the most significant cause of deforestation, governance and legal aspects, and capacity building.

The researchers identified any problems within the problem trees that are capacity issues. Note that although the problem trees are designed to identify the underlying causes, for this analysis, capacity problems identified at any level of the problem tree were included because REDD+ activities are likely to address capacity issues at many different 'problem levels'. The problems at the bottom of the tree are the key proximal causes, and addressing these are likely necessary to address other issues higher in the tree, and the discussion will consider how well the REDD+ activities address these underlying issues. However, REDD+ activities might address problems at other levels, and so all are included here.

REDD+ capacity building activities were taken directly from DRC's REDD+ National Strategy [40], translated from French by the researchers. The strategy includes 'Tables of Investment Cycles for REDD+' divided by sectors, with activities classified into seven categories, with one being Capacity Building (Renforcement des capacités). Any activity that was considered as capacity building in the REDD+ strategy was included in the analysis to ensure all capacity building was examined.

In addition, the workshop data were supplemented by results from a community survey about the governance and planning of REDD+ and sustainable development activities carried out in two rural communities in the Équateur province, Buya 1 and Bongonde Drapeau, and in the provincial capital Mbandaka. The villages were the focus of REDD+ efforts as part of Projet Équateur coordinated by the Woodwell Climate Research Center, and the surveys were part of the work. The surveys asked local community members to rate 15 indicators of governance and planning and give comments on how they could be improved. The survey had 157 respondents (demographic details of the respondents are provided in the Supplementary Information, Tables S1–S4), all of whom provided short free text comments on the 15 indicators. The majority of the respondents were local farmers or forest users, but some government and NGO officials were also included. The results from all respondents are used here because capacity building affects all sectors. The comments were thematically coded in NVivo to identify common themes and trends, some of which aligned with the capacity issues highlighted in the workshops. In the results, for each workshop 'problem', the overall number of comments that are coded to the related code and the number of respondents that make comments that include the related code are reported (any particular code may have been mentioned more than once by a respondent). The aim of the survey was not to identify capacity needs specifically, but many responses in the free text part of the survey referred to capacity needs. Hence, the survey data provides some insight into community capacity needs, particularly in the context of governance and planning, but may not reflect the full picture.

### 2.2. Data Analysis

An analytical framework was created to allow for comparison of the capacity building goals of the DRC REDD+ Plan with those of provincial and local stakeholders. The analytical framework has two axes that look at (1) different levels of capacity building and (2) different types of capacity (see Supplementary Information, Table S5, for descriptions of levels and types, and Table S6 for examples).

Capacity building is often focused at different levels or scales. Capacity building might be aimed at individuals or organisations, as highlighted by the UNFCCC [24], through, for example, training or education. Sectoral or institutional capacity building [32] is aimed at the institutions in a sector or more broadly across government and civil society. Systemic capacity building [24] is focused on enhancing the 'enabling environment' capacity [32] for institutions to take actions, such as developing strong governance systems. Note that there is likely to be overlap between these scales, and they are all likely to include some element of training or education at an individual level. Similarly, building community capacity might be considered the organisational level, but may focus on community-scale institutions.

Capacity building will also have specific goals in mind. Building capacity at any level can focus on technical capacity building focuses on key areas of expertise around an issue, or functional capacity building that focuses on the management and organisational skills required for effective functioning of projects, programs and policy implementation [32]. Participative capacity building seeks to enable stakeholders to be involved in decision-making, policy and planning processes [53]. Integrative capacity is focused on improving the way people work together, while strategic capacity looks to build long-term, holistic problem-solving [53]. Finally, importantly in the context of climate change, adaptive capacity building focuses on enhancing the "ability of a system to adjust to change, to take advantage of opportunities, or to cope with the consequences" [32]. As with the different scales, these types of capacity building are likely to overlap—many capacity building measures within REDD+, for example, are focused on involving stakeholders in technical activities such as forest monitoring, which could build both technical and participative capacity. Note that defining scale and type of capacity are not designed to be rigid definitions, but provide a framework for comparing the alignment of capacity building needs and activities—in this case between the needs identified by stakeholders and the activities defined in a national REDD+ strategy. In using this framework, the data analysis consisted of two parts (Figure 1).

1. We classified the scale and type of capacity needs identified by the problems in the problem tree and activities in the REDD+ strategy according to the framework discussed above (see Supplementary Information, Table S7, for full analysis). Note that any awareness raising activity was classified as participative capacity building, even though it is considered low-level participation [54].

2. We compared the capacity needs and capacity building activities by:

    a. Identifying activities from the strategy that are likely to address the problems identified by the stakeholders in the problem tree;

    b. Comparing overlaps and mismatches between the types and scales of capacity needs and capacity building activities identified.

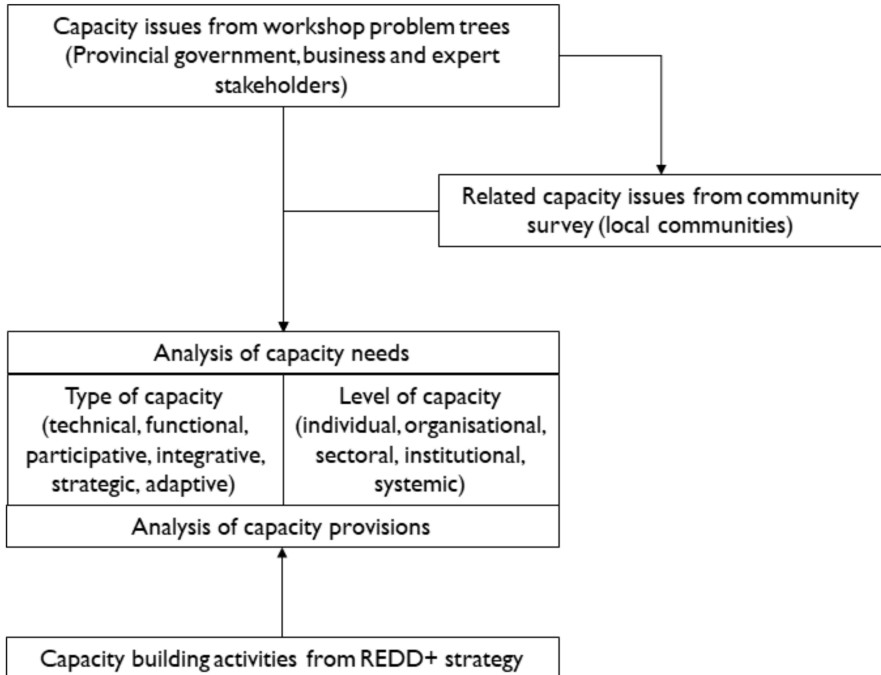

**Figure 1.** Analytical framework for comparing capacity needs identified by government and expert (workshop) and community (survey) stakeholders with capacity building activities identified in the DRC REDD+ strategy.

### 3. Results: Capacity Building Needs to Combat Deforestation and for REDD+

The sectoral problem trees are shown in the Supplementary Information, with problems related to capacity building highlighted in orange boxes. The capacity building activities for REDD+ are taken directly from the national REDD+ strategy and are presented in Supplementary Information, with each activity assigned scale(s) and type(s) of capacity.

In terms of type of capacity, the majority of the problems identified by stakeholders require technical and functional capacity building, but the issues also suggest the need for integrative, participatory and strategic capacity. The strategy appears to have a greater focus on participatory capacity, likely due to many activities focused on awareness raising and other participation activities (Figure 2). In terms of scale, the capacity building needed to address the workshop problems and the capacity building activities in the strategy show a similar distribution. Note, however, that these similarities in distribution do not mean that the strategy necessarily aligns with the needs identified by the stakeholders—what is more important is how individual activities align with the problems identified by the stakeholders.

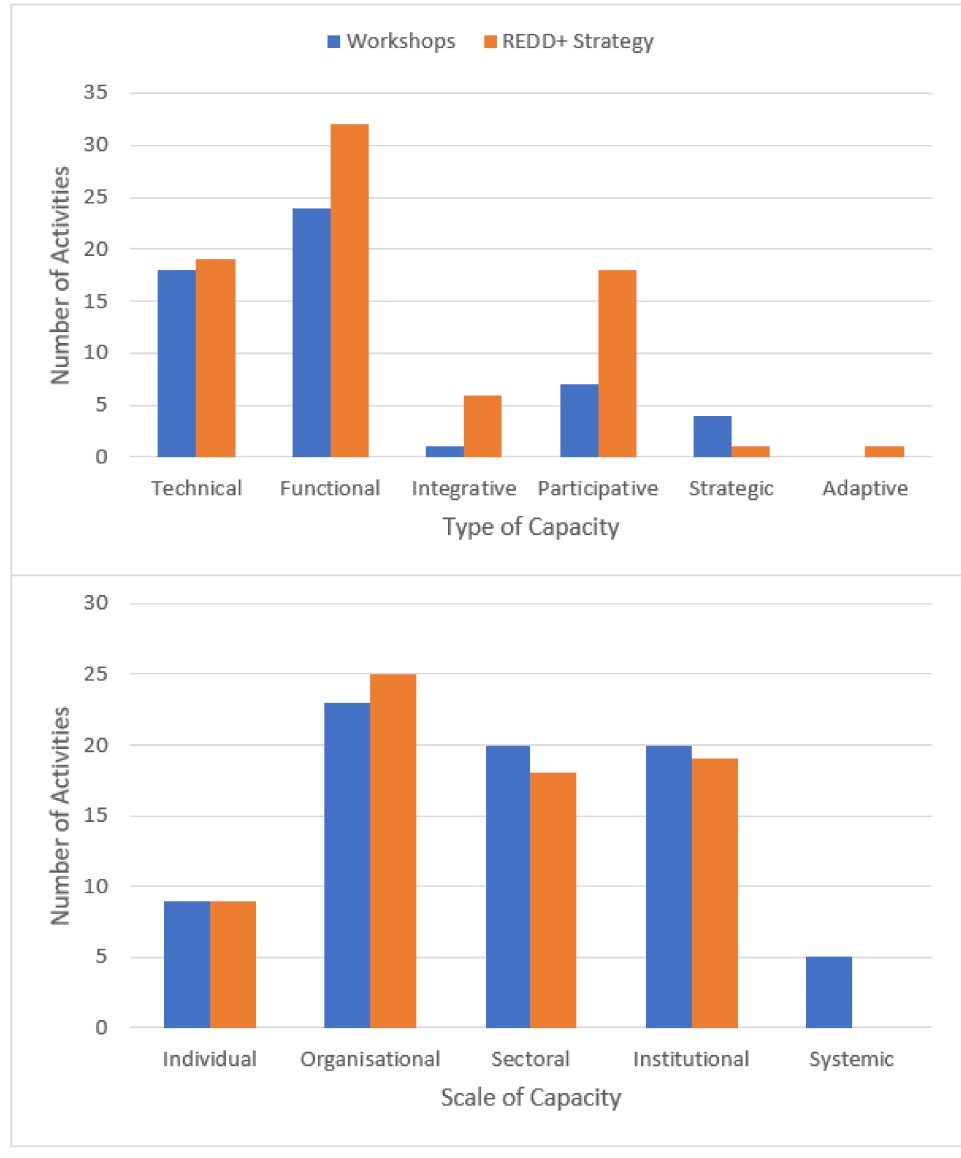

**Figure 2.** Different types and scales of capacity building activities identified in stakeholder workshops and the DRC REDD+ Strategy. Note: activities might include more than one type or scale of capacity, and so numbers of activities are not directly comparable.

In each section, the problem tree issues that are related to capacity building on a sector-by-sector basis are shown in a table, with the relevant REDD+ activities that align to them presented on the right-hand side of each table. Issues from the bottom of the tree are highlighted as these are taken to be underlying causes; however, other problems related to capacity are also included.

*3.1. Agriculture*

Both the local workshop and the REDD+ strategy recognise that current shifting agricultural practices are a major contributor to deforestation, and that changes to this sector are essential and most urgent, in line with recent major studies [8,14,35]. Hence, forest protection and restoration efforts supported by REDD+ must address deforestation caused by shifting agriculture while ensuring food security. REDD+ can be used to provide alternative sources of income through carbon markets, but food security concerns mean it may not be possible to simply replace shifting agriculture; however, there may be opportunities to prevent further primary forest loss or to fund restoration [55]. Improving agricultural practice to increase yields and reduce food security is a key focus of RECC+ projects in the Équateur province [13].

In the Équateur province, the workshop stakeholders noted the need for local level research capacity (row 1a, 1b, 1c, Table 1), both in terms of technical knowledge and functional capacity of local research organisations. Similarly, the need for education and training around key problems was a common issue mentioned in the survey (55 mentions by 47 respondents). Hence, any research needs to ensure that it helps support the community in understanding and responding to issues, i.e., research with effective communication. There is significant overlap with the REDD+ strategy on these technical capacity issues. The strategy highlights the need for research and supporting technical services in order to communicate that knowledge. However, it is not clear from the strategy how much will be invested in research or where. The Investment Plan also highlights agricultural research as a key area for investment, but whether there will be investment in the Équateur province and what this will look like is unclear. Many of the existing initiatives around research in the Congo are funded through international research grants implemented by a wider range of global academic and applied government and non-government organisations. Local technical capacity exists—there are DRC national and province level tertiary education establishments, and one of the three national botanical gardens (Jardin Botanique d'Eala) in the main city of Mbandaka, who are already on-ground partners for Projet Équateur. However, these provincial organisations have extremely limited technical, financial, and administrative capability; indeed, they themselves need support to develop general operational capacity development, exacerbating the challenge to build a durable approach to deliver specialist training capabilities to support the focus on REDD+ initiatives.

The absence of policy and limited funding for the agricultural sector was another key issue highlighted by the workshop stakeholders (row 1d, 1e, Table 1). The need for agricultural financial support was a common theme within the community survey responses (62 mentions by 52 respondents). The REDD+ strategy includes support for building this type of functional and strategic capacity through agricultural initiatives and support for experimenting with financial arrangements, which could provide funding to the agricultural sector. However, as the workshop stakeholders identified, there is a broader strategic need for an agricultural policy for the province. Hence, there is a challenging mismatch: the province needs to show that they have an agricultural policy that will combat deforestation in order to justify and secure REDD+ resources and investment in the region, but capacity at the provincial level is limited.

**Table 1.** Agricultural Sector Capacity Issues Identified in Équateur Provincial Government Stakeholder Workshop and Aligned DRC REDD+ Strategy Actions. Highlighted issues in grey are proximal issues from the bottom of problem tree issues.

| Agriculture | | | | | | |
|---|---|---|---|---|---|---|
| **Issue** | **Capacity** | | **Related Survey Codes (Number of Mentions; Number of Respondents)** | **Alignment to Activities in DRC REDD+ Strategy** | **Capacity** | |
| | **Type** | **Scale** | | | **Type** | **Scale** |
| 1a Absence of local research | Technical | Organisational Sectoral | Research (11; 10) | 1.15. Support research for the development of technical reference systems for sustainable agriculture, on the links between agriculture and forestry | Technical | Organisational Sectoral |
| | | | | 1.18. Create model farms with a centre for rural innovation and plant propagation to support zoning plans | Technical Adaptive | Organisational Sectoral |
| 1b Local research organisations (INERA and CAPS) non-functional | Technical Functional | Organisational | — | 1.16. Renewing technical services and building their capacity on sustainable agriculture technologies | Technical | Organisational |
| 1c Lack of funding for applied research | Technical Functional | Organisational Sectoral | Research (11; 10) Awareness of problems and issues (47; 60) | 1.15. Support research for the development of technical reference systems for sustainable agriculture, on the links between agriculture and forestry | Technical | Organisational Sectoral |
| | | | | 1.18. Create model farms with a centre for rural innovation and plant propagation to support zoning plans | Technical Adaptive | Organisational Sectoral |
| 1d Lack of funding available for the agricultural sector | Functional | Sectoral | Financial support (62; 52) | 1.14. Experimenting with various financial instruments to support agricultural operators involved in a REDD+ approach | Functional | Organisational Sectoral |
| 1e Absence of agricultural policy in the province | Strategic | Institutional | — | — | — | — |
| 2a Lack of information and training | Technical | Organisational Sectoral | Agricultural training (55; 47) | 1.3. Promote participation in sustainable production roundtables | Technical | Individual Organisational |
| | | | | 1.16. Renewing technical services and building their capacity on sustainable agriculture technologies | Technical | Organisational |
| | | | | 1.18. Create model farms with a centre for rural innovation and plant propagation to support zoning plans | Technical Adaptive | Individual Organisational |

**Table 1.** *Cont.*

| | | Agriculture | | | | | |
|---|---|---|---|---|---|---|---|
| | **Issue** | **Capacity** | | **Related Survey Codes (Number of Mentions; Number of Respondents)** | **Alignment to Activities in DRC REDD+ Strategy** | **Capacity** | |
| | | **Type** | **Scale** | | | **Type** | **Scale** |
| 2b | Low awareness of agricultural laws | Participative | Individual Sectoral Institutional | Awareness of existing laws and regulations (12; 10) | 1.3. Promote participation in sustainable production roundtables (Agriculture) | Technical Participative | Individual Organisational |
| | | | | | 1.6. Support and strengthen the capacities of steering and consultation frameworks at the higher levels of local governance (Groups, Sectors, Territories) | Functional Participative | Organisational Institutional |
| 2c | State services not fulfilling their mission | Functional | Organisational Sectoral Institutional | — | 1.16. Renewing technical services and building their capacity on sustainable agriculture technologies | Technical | Organisational |
| 2d | Lack of motivation from civil servants due to unpaid salaries | Functional | Institutional | — | — | — | — |
| 3a | Absence of agricultural input providers | Functional | Sectoral | Agricultural inputs (68; 50) | 1.7. To support any form of initiative related to sustainable agriculture aimed at organizing stakeholders in the commodity chains | Technical Functional Participative | Organisational Sectoral |
| | | | | | 1.13. Supporting the emergence of financial services in rural areas in support of sustainable agricultural practices | Functional | Organisational Sectoral |
| 4a | Lack of government means | Functional | Institutional | Strengthen state capacity (1; 1) | 1.6. Support and strengthen the capacities of steering and consultation frameworks at the higher levels of local governance (Groups, Sectors, Territories) | Functional Participative | Organisational Institutional |

Lack of agricultural information, extension and training was an issue that the REDD+ Strategy has identified and has several activities that align well with (row 2a, Table 1). However, poor awareness was identified as an underlying factor during the workshop (row 2a, Table 1) and was occasionally mentioned in the community surveys. Raising awareness, especially among communities, requires participative capacity building. The REDD+ strategy does include some elements of this type of capacity, such as promoting participation in and the capacity of agricultural groups, although these tend to be more focused on technical capacity at the organisational scale with limited focus on vocational training (row 2b, Table 1).

However, in the workshop, the problem tree identified that the key capacity issue for a lack of information and awareness was limited government capacity (row 2c, 2d, Table 1). The lack of government capacity is a common proximal issue within the problem trees (e.g., lack of government means), especially the capacity building problem tree (see below). The REDD+ strategy does recognise the need to strengthen state capacity (row 4a, Table 1), although this is likely to be focused on REDD+ technical capacity specifically, whereas the issues extend beyond simply the capacity to carry out REDD+. Other capacity issues fall beyond the scope of the REDD+ strategy. Recently, provincial governments have begun receiving taxes directly instead of through the national government, so there is at least the potential for more resourcing for policy development and implementation; however, this also relies on the necessary technical and functional capacity. In addition, even if governments receive further funding, ensuring that communities receive the financial (and other) support identified by the survey respondents is a significant challenge, related to governance issues (see below).

### 3.2. Governance and Legal Aspects

As noted above, some of the agricultural capacity issues identified are governance-related issues. The governance and legal aspects problem tree developed in the workshop reflect many similar issues, although focus on a more general level. A lack of technical and functional capacity within the civil service around sustainable forest management and more generally was a key problem, related to the weak provincial government sector (row 1a, 1b, 1c, 1d, 3a, 3b, Table 2). Notably, the general lack of education, training and awareness highlighted in the workshop was also reflected in the community survey, where the need for training, education and awareness raising were common themes that arose in 273 comments from 126 users. Similarly, a lack of awareness of laws in the general population was highlighted (row 2a, 2b, Table 2), which is classified here as a participative capacity issue. The community also raised this problem occasionally (the need for greater awareness of existing laws and regulations was noted 12 times by 10 participants).

**Table 2.** Governance and Legal Capacity Issues Identified in Équateur Provincial Government Stakeholder Workshop and Aligned DRC REDD+ Strategy Actions. Highlighted issues in grey are proximal issues from the bottom of problem tree issues.

| Governance | | | | | | | |
|---|---|---|---|---|---|---|---|
| | **Problem** | **Capacity** | | **Relevant Survey Codes (Number of Mentions; Number of Respondents)** | **Alignment to Activities in DRC REDD+ Strategy** | **Capacity** | |
| | | **Type** | **Scale** | | | **Type** | **Scale** |
| 1a | Lack of capacity of some of the civil servants at all levels of administration | Technical Functional | Organisational | Activities and support: Methods: Governance activities: Strengthen state capacity (1; 1) | 4.14. Strengthen the implementation of national legal instruments and popularize them | Functional Technical | Institutional |
| 1b | Lack of awareness of sustainable forest management | Technical Participative | Organisational Sectoral | Activities and support: Support needed: Awareness raising: Awareness of sustainability (4;4) | 3.A.18. Raise awareness of sustainable forest management practices and build local capacity to support communities towards the sustainable management of their forest resources | Technical Participative | Organisational |
| | | | | Activities and support: Support needed: Awareness raising: Awareness of problems and issues (e.g., deforestation) (60; 47) | 3.B.5. Supporting community-based conservation and sustainable collaborative natural resource management | Functional Participative | Organisational Sectoral |
| 1c | Lack of training program for civil servant | Technical Functional | Organisational | — | 4.2. Test, improve, and use online tools for cross-monitoring REDD+ implementation and impact: interconnection of the SNSF, the National REDD+ Registry, and the Moabi | Technical Integrative | Organisational Sectoral |
| | | | | | 4.7. Strengthen the capacity of stakeholders to play their roles in planning, implementing and monitoring REDD+ | Technical Functional | Individual Organisational |
| 1d | Absence of systematic training for new legislation | Technical | Organisational Institutional | Activities and support: Support needed: Awareness raising: Awareness of existing laws and regulations (12; 10) | 4.14. Strengthen the implementation of national legal instruments and popularize them | Technical Functional Participative | Institutional |
| 2a | Ignorance of the law by part of the population | Technical Participative | Individual Organisational | Governance issues: Existing laws and regulations (19; 19) | 7.13. Popularize the land law and build the capacities of consultation frameworks and various stakeholders | Participative Functional | Organisational |

**Table 2.** *Cont.*

| | | | Governance | | | |
|---|---|---|---|---|---|---|
| **Problem** | **Capacity** | | **Relevant Survey Codes (Number of Mentions; Number of Respondents)** | **Alignment to Activities in DRC REDD+ Strategy** | **Capacity** | |
| | **Type** | **Scale** | | | **Type** | **Scale** |
| 2b  Absence of training, education and information | Technical | Individual Organisational | Support needed: Awareness raising (273; 126) Support needed: Training (73; 61), education (6; 5) | 4.10. Support the emergence and capacity building of national service companies in support of REDD+ | Functional | Organisational |
| 2c  Lack of facilitators/guidance for facilitators | Functional | Organisational Sectoral Institutional | Methods: Use community liaisons or facilitators (5; 5) | 4.12. Implement monitoring mechanisms on the ground to deal with complaints received and compliance with national standards | Functional | Institutional |
| 3a  Weak provincial public sector | Functional | Institutional | Methods: Governance activities: Strengthen the authority of the state (4; 3) | — | — | — |
| 3b  Absence of mobile agents | Functional | Sectoral Institutional | Methods: Engagement: Community involvement in project activities (implementation) (125;80) | — | — | — |

The REDD+ strategy does include activities focused on technical and functional capacities within the civil service and government at all levels, as well as some focused on participative capacity designed to increase awareness of the public. However, many of these are focused specifically at technical and functional capacity for governance and legal aspects of REDD+, rather than the more general skills the stakeholders and the community participants focus on. This suggests there might be a mismatch between the focus of the REDD+ strategy and the need for training, awareness and education identified by stakeholders, both in terms of the scale and type of knowledge. The strategy focuses on government rather than community, also supported by the fact that workshop stakeholders noted the need for agents in the field to support the community, something reflected in the survey by the desire of the respondents to be involved in activities (125 comments from 80 participants). The strategy also focuses on functional rather than technical knowledge; but the results here suggest that both are needed, with technical knowledge important for the community, especially around agriculture (55 comments from 47 respondents) and key issues including sustainability. Community respondents are focused on activities that improve livelihoods, given the level of poverty, so are seeking direct training and awareness raising in activities that will improve incomes, food security, or address other development issues.

### 3.3. Capacity Building

The final problem tree developed by the stakeholders concerned capacity building directly. Note that this tree was not revised in 2019 due to time constraints, but discussion with the stakeholders revealed that the same issues persisted. The problem tree emphasises a lack of technical, functional and participative capacity. The stakeholders felt that this was largely because many of the ideas around REDD+ and the sustainable development that underpins it are very new to decision makers, policymakers and communities (row 2a, 3c, 5, Table 3). The community also highlighted the need for an improved knowledgebase and greater awareness around issues. Awareness raising generally (which covered a number of subjects, including awareness around issues, awareness of existing activities and regulations) was mentioned 273 times across 126 participants. More specifically, the need for greater awareness around problems and issues related to deforestation was common (60 comments across 47 participants). Notably, sustainability specifically only appeared four times across the survey, but this likely suggests that communities are not necessarily aware of the specific jargon—issues around sustainable development were common. In addition to awareness raising, training was a common capacity need identified (73; 61), usually focused on agricultural training but also other activities, including management and participatory approaches. The community were acutely aware of the need for greater education (awareness raising) and training.

This implies a systemic capacity challenge that needs to be addressed. Stakeholders at all levels need to understand the importance of a sustainable development approach, the importance of participation in this, and the skills to share and implement that understanding. However, research in the province suggests that knowledge and teaching of sustainable development and environmental issues is limited within the education system [51].

Finally, the workshop identified the need for more participatory approaches (row 3a, 3b, Table 3). Similarly, community survey respondents strongly highlighted the need for participatory approaches, especially around consultation and involvement in activities, with the need for more or better participatory consultation mentioned 152 times across 99 participants and participation concerns and issues being mentioned 84 times by 63 respondents. This aligns with the REDD+ strategy, which seeks to support community capacity and governance. However, the workshop participants felt that the limited knowledge base around fundamental issues of sustainable development (row 3, Table 3) was a major barrier to these participatory approaches.

**Table 3.** Capacity Building Issues Identified in Équateur Provincial Government Stakeholder Workshop and Aligned DRC REDD+ Strategy Actions. Highlighted issues in grey are proximal issues from the bottom of problem tree issues.

| | | Capacity | | Relevant Survey Coding (Number of Mentions; Number of Respondents) | Alignment to Activities in DRC REDD+ Strategy | Capacity | |
|---|---|---|---|---|---|---|---|
| | **Problem** | **Type** | **Scale** | | | **Type** | **Scale** |
| 1a | Insufficient training and information about deforestation and sustainable development | Technical | Individual | Support needed: Awareness raising: Awareness of problems and issues (60; 47) | — | — | — |
| 2a | New subject: Sustainable development and environmental issues are a 'new subject' to many | Technical | Individual Organisational Sectoral Institutional Systemic | Support needed: Awareness raising: Awareness of problems and issues (60; 47) Support needed: Awareness raising: Awareness of sustainability (4; 4) | 1.3. Promote participation in sustainable production roundtables (Agriculture) | Technical Participative | Individual Organisational |
| 3a | Lack of a participatory approach | Participative | Systemic Institutional Sectoral | Methods: Engagement: Participatory consultation (152; 99) Governance issues: Participation concerns and issues (84; 63) | 3.B.6. Supporting the structuring of local communities and indigenous peoples and strengthening their capacities in the long term | Functional | Institutional |
| 3b | Lack of trainers in these issues (including participation) | Functional Participative | Individual Organisational Sectoral Institutional | Support needed: Training, coaching (73; 61) Support needed: Training, coaching: Training-community participation (2; 2) | — | — | — |
| 3c | New subject: Sustainable development and environmental issues are a 'new subject' to many | Technical | Individual Organisational Sectoral Institutional Systemic | Support needed: Awareness raising: Awareness of problems and issues (60; 47) Support needed: Awareness raising: Awareness of sustainability (4; 4) | 1.3. Promote participation in sustainable production roundtables (Agriculture) | Technical Participative | Individual Organisational |
| 4a | Lack of information in local language | Functional | Organisational | Governance issues: Participation concerns and issues: Language barrier (1;1) | — | — | — |
| 5 | New subject: Sustainable development and environmental issues are a 'new subject' to many | Technical | Individual Organisational Sectoral Institutional Systemic | Support needed: Awareness raising: Awareness of problems and issues (60; 47) Support needed: Awareness raising: Awareness of sustainability (4; 4) | 1.3. Promote participation in sustainable production roundtables (Agriculture) | Technical Participative | Individual Organisational |

The province has recently developed a sustainable development plan, suggesting that capacity in sustainable development has increased since 2016, at least within the provincial government. Nonetheless, since sustainable development underpins REDD+, its purpose and its implementation, this lack of capacity has the potential to be a significant barrier to effective action.

## 4. Discussion

The analysis of capacity building needs identified by provincial government and comparison with the REDD+ strategy stakeholders identifies a number of issues.

The stakeholders and the REDD+ strategy both identify a broad range of types and scales of capacity building required to address issues. Both focused significantly on technical and functional capacity. From a REDD+ perspective, this is unsurprising because implementing REDD+ requires a high degree of technical and functional capability to ensure integrity and demonstrate transparency. The stakeholder focus on functional capacity reflects the institutional challenges of the DRC bureaucracy [19,44]. Their focus on technical capacity reflects similar challenges in developing countries worldwide—access to highly technical education, training and resources more generally is highly limited by resources.

The REDD+ strategy had a number of activities that aimed at building participative and integrative capacity, the provincial government stakeholders recognised the importance of participation in the 'capacity building' problem tree, and participation was a common them in the community survey. The REDD+ strategy was more explicit in highlighting that activities should be (or could be) participatory, likely reflecting the importance placed on participation in UNFCCC REDD+ processes and procedures. The REDD+ strategy also had a number of consultation or awareness-raising activities that were classified here as participative capacity building. However, it is important to note that these were always low-level participation activities [54].

In some areas, there was strong overlap between the capacity needs identified by stakeholders and the capacity building activities in the REDD+ strategy. Technical capacity building needs in the agricultural sector were highly aligned between both stakeholders and the strategy. This no doubt reflects the fact that much of the DRC are facing the same challenges around deforestation, and so Équateur's issues are reflected in the strategy. The data here suggest that the REDD+ strategy is largely 'on the right track' to address the major cause of deforestation in the Équateur province. At the same time, the results show there are many opportunities for the provincial government to harness REDD+ funds to address the capacity needs around deforestation. Further work is needed to more specifically identify where these opportunities are, thus helping the government target their requests for funds to prioritise areas that are most likely to be successful.

There are, however, a few areas where there are mismatches of either the type or scale of the capacity needs identified in Équateur and the activities in the strategy. In some areas, especially governance and policy, provincial government stakeholders identified problems that require high level systemic or institutional capacity building to address some fundamental capacity problems. The governance problem tree compared to the governance activities in the REDD+ strategy highlight this mismatch, with the REDD+ strategy focusing on specific elements of REDD+ governance that are required. Although this is, in part, due to the broader aims of the REDD+ strategy and the more specific problem, deforestation, the stakeholders were asked about, since the aim of REDD+ is ultimately to prevent deforestation, these mismatches are a cause for concern.

Importantly, some of the issues raised by stakeholders are a function of the governance and political challenges that are common across the DRC and systemic. REDD+ is not going to be sufficient or able to address these issues, as the REDD+ investment strategy makes clear: "The scale of these programmes alone is insufficient to address fundamental underlying drivers (policies, legal framework, etc.), which constitutes severe limitation on what project and jurisdictional types of approaches can achieve" [39]. Hence, the more

institutional and systemic capacity needs identified by the stakeholders are not likely to be directly addressed by REDD+ alone.

Ultimately, the mismatches highlight the risks for implementation of REDD+. If other capacity is not in place, then REDD+ activities are at high risk of failing. For example, a lack of understanding of the issues of sustainable development highlighted by this research and elsewhere is potentially a fundamental barrier to REDD+ implementation. If stakeholders are not aware of what, ultimately, REDD+ is seeking to achieve, implementation will be difficult. The new agreement on carbon markets under Article 6 of the Paris Agreement may potentially add further challenges; it focuses heavily on environmental integrity [23], another concept that stakeholders will need to understand. While the new rules recognise the need for capacity building, this is likely to focus on technical and functional capacity to implement the rules, without the more systemic capacity needed to achieve the desired outcomes.

Similarly, in the context of Équateur province, there is a lack of some fundamental capacities, such as simply knowledge of key issues in sustainable development more broadly. This includes a lack of education on environmental and sustainability issues at the university level, and the training and information that is available is of limited quality [51]. Knowledge and training around policy, management and governance is also missing, and risk being fundamental barriers that will simply prevent any REDD+ effort.

Ultimately, this highlights that REDD+ cannot be a panacea to address deforestation or other more systemic problems in the DRC or elsewhere. There is a hope implicit within REDD+, and sometimes explicitly stated, that building the specific capacities for REDD+ will help address these broader governance, policy, and planning capacity issues. Certainly, good technical and functional capacity at the individual, organisational and sectoral level can help build systemic, institutional capacity, but it is far from certain. If this is the hope for REDD+, there is the need for research and monitoring that looks for evidence that it is having an impact at the higher levels and to identify what sorts of projects are likely to have this impact. In the DRC context, systemic capacity building—education and awareness of sustainability issues, governance, and good management—are needed alongside the technical capacity if REDD+ is to be successful.

Finally, this research adds further weight to the need for the local context to be considered. The risks discussed above, caused by the capacity mismatches highlighted here, can, to some extent, be mitigated by a good understanding of the local context. Understanding what formal and, perhaps more importantly in the DRC context, informal governance arrangements are in place can help mitigate the risks caused by missing capacity. There is no question that DRCs reputation as a 'fragile state' [19]—the lack of political will and/or capacity to provide the basic functions needed for poverty reduction, development [47]—makes building capacity for functioning REDD+, or other projects, more complex. Nonetheless, experience on the ground suggests that capacity and governance can be built and there are actors with capacity and willingness to learn and take part, especially at the local level. Furthermore, we suggest that effectively building capacity that 'feeds upwards' requires long-term projects that are as embedded as possible in the local context and focus on training local people and building local relationships. This research is part of the ongoing Projet Équateur (Zamba Malamu), which has been operational on the ground in Équateur Province since 2013. The motivation for the project was to address the immediate challenges for REDD+ preparation, moving to develop capacity for delivery. Through experience, the project identified the more systemic issues affecting the underlying institutional and organisational capacity to deliver any form of development, and began to use REDD+ as a lens to focus on the near-term pathways to implement action on the ground in a "failed state".

## 5. Conclusions

Addressing deforestation and forest degradation caused by smallholder shifting agriculture has been identified as the key priority in the DRC. Implanting the REDD+ national

strategy as presently conceived holds enormous potential to support primary forest protection, especially in highly forested, low deforestation developing countries such as the DRC; however, its potential is stifled by the present weak organisational and technical capacity to plan, manage and mobilise funds to address the key drivers of deforestation.

The REDD+ investment strategy recognises some of the needs for organisational capacity building to address key weaknesses and challenges in policy, governance and management, and these fundamental issues are also identified by the provincial stakeholders. Present investment priorities are targeted at the frontline actions to develop the framework for the national forest carbon market. Current action focuses on technical and participative capacity at a sectoral level to change agricultural practices, alongside other technical capacity for monitoring and evaluation of forests to meet the stringent MRV requirements of REDD+.

However, scaling impact will be impossible without the human capacity to manage the system. The REDD+ strategy is limited in how much it can address some of these issues that stem from political instability and lack of political will, and it recognises this. Greater awareness of, and capacity to address, these issues is needed to ensure the sustainable management of forests. This research shows that both the local stakeholders and the community are seeking knowledge and training to address these issues, and capacity building is a key part of this. The next phase of national carbon market development needs to prioritise resolving systemic capacity building bottlenecks. Education and training in policy, management, governance, and sustainability more generally needs to be an important part of REDD+. Such systemic capacity building calls for durable strategy to invest in higher education and vocational training, and which recognises and understands the local context. Current and future REDD+ projects in the DRC risk failure because although they build structures and processes, if capacity building is ignored, there are no people to populate those structures.

**Supplementary Materials:** The following supporting information can be downloaded at: https://www.mdpi.com/article/10.3390/land11060918/s1, Tables S1–S4: Demographic details of survey respondents; Table S5: Levels and types of capacity building; Table S6: Examples of scales and types of capacity building, with examples relevant to REDD+; Figure S1: Problem tree for the agricultural sector; Figure S2: Forest loss and degradation problem tree for governance and legal aspects; Figure S3: Forest loss and degradation problem tree for capacity building; Figure S4: Forest loss and degradation problem tree for biomass sector; Figure S5: Forest loss and degradation problem tree for forestry sector; Table S7: Capacity type and scale assessment of REDD+ capacity building activities.

**Author Contributions:** Conceptualization, E.A.M., G.B. and J.Z.M.; methodology, E.A.M. and G.B.; formal analysis, E.A.M.; investigation, E.A.M., G.B., J.Z.M. and M.K.; resources, G.B. and J.Z.M.; data collection: E.A.M., G.B., J.Z.M. and M.K.; data curation, E.A.M. and M.K.; writing—original draft preparation, E.A.M.; writing—review and editing, G.B. and J.Z.M.; project administration, G.B. and J.Z.M.; funding acquisition, G.B. and B.M. All authors have read and agreed to the published version of the manuscript.

**Funding:** This research was funded by a charitable organization which neither seeks nor permits publicity for its efforts. The trust has had no influence on the design, analysis, interpretation and documentation of this research.

**Institutional Review Board Statement:** The study was conducted in accordance with the Declaration of Helsinki and in line with the Australian National Statement on Ethical Conduct in Human Research (2018), and approved by the Ethics Committee of Griffith University (protocol code 2019/762, Approval Date: 10 September 2019).

**Informed Consent Statement:** Informed consent was obtained from all subjects involved in the study.

**Data Availability Statement:** The data presented in this study are available on request from the corresponding author. The data are not publicly available due to ethical constraints.

**Acknowledgments:** The authors would like to thank all the community members and government officials who took part in the workshops and surveys.

**Conflicts of Interest:** The authors declare no conflict of interest. The funders had no role in the design of the study; in the collection, analyses, or interpretation of data; in the writing of the manuscript, or in the decision to publish the results.

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
