# Peer review of "Comparing Community Needs and REDD+ Activities for Capacity Building and Forest Protection in the Équateur Province of the Democratic Republic of Congo"

_land, doi:10.3390/land11060918_

Round 1

Reviewer 1 Report

The article is very interesting, is well written and fits the aim of the journal. It will be of interest to the readers.

In my opinion some issues can be improved:

  1. Usually, I don't like to see a section indication, e.g., 1. Introduction, followed by text, and after a subsection, e.g., 1.1. REDD+, Forest Protection and Capacity Building. The authors, instead of this, should use 1.1. Framework, immediatly after 1. Introduction. The same for all similar situations.
  2. The authors should use "1.1." and not "1.1". Use always a dot after the number.
  3. Section 1 must include a paragraph explaining the objectives of the work.
  4. Must improve the quality of Figure 1.
  5. Section 3 - see comments in point 1.
  6. Must improve quality of Figure 2 and do not frame the pictures. Avoid 3D graphs.
  7. Table 1, 2 and 3 are too far from the explanation text.

Author Response

Thank you for the positive comments. Please find attached a table with responses to each comment. Please note that we have not been able to make some changes due to the use of the MDPI template.

Author Response

Thank you for your positive and constructive comments, which have helped improve the manuscript. Please see the attachment for responses to your specific points.

Reviewer 3 Report

Democratic Republic of Congo DRC is a very reach country in what concerns natural resources. The problem of the country is the lack of commitment on the part of Governments to implement policies to defend the country's immense natural wealth, some of them of key value in world markets.

The study does a conjunctural analysis unacceptably decoupled from the DRC structural problems. Therefore, the study can’t have any effect on the structural problems of forest management either locally or at country level

Author Response

Thank you for your comments on the paper.

We acknowledge that there are deep political and structural issues in the DRC. Although, we do not go into the details of this we do discuss them, and have improved the discussion of them Lines 156-187.

Certainly, the aim of this paper is not to “effect the structural problems of the country”. We suggest that would be beyond the scope of any academic paper. We also do not think there is value in repeating the analyses of these problems – others have done that far better already.

Nonetheless, we place this paper firmly in that context, and suggest that capacity building, and importantly the right sort of capacity building, is a key base process in this context. We think there is value in providing an empirical analysis of this in the context of REDD+ and forest management.

Reviewer 4 Report

This study explores how the capacity building activities within the DRCs REDD+ strategy are aligned with the capacity needs identified through workshops and surveys. Although there are still many systemic capacity needs that are unlikely to be addressed, many technical capacity needs identified by stakeholders could be potentially addressed by the REDD+ strategy. Failure to address these needs causes risks that would undermine any implementation of REDD+. The results suggest that education and training in governance and management, as well as fundamental education in sustainability are key capacity needs. Overall, this study addresses an important issue and the paper is well organized. However, some details need to be elaborated to improve the readability. Detailed comments are shown as follows.

  1. In the introduction, progress on deforestation and forest degradation is suggested to be provided. Information that covers the deforestation condition in Congo (Lines 114-146) could be moved to the introduction.
  2. It is suggested to add references for the statement "forest cover in DRC, estimated around 105 million ha in 2000, with 60% classified as within 'intact forest landscapes." in Lines 116-118.
  3. The details on the implementation of REDD+ policies in DRC should be provided in Section 1.2. Also, please add elaborations for the deforestation and degradation condition in DRC in Section 1.
  4. It is suggested to provide more information about the challenges in implementing REDD+ policies in DRC in Lines 156-165.
  5. Please provide an explanation about why the two communities are selected in this study and how the survey and sampling process are conducted. It is also important to provide background information about the 157 respondents in Section 3.

Author Response

Thank you for the positive and constructive comments on our paper. Please see the attachment for specific responses and details of changes made.

Round 2

Reviewer 3 Report

The study does a conjunctural analysis unacceptably decoupled from the DRC structural problems.  

The matter of structural problems of DRC in the protection of natural resources must be patent in the Introduction and Conclusion using synthetic and explicit wording and references shall be provided. 

The paper shall raise awareness in the participants, intermediate and end-users. Conveying the idea that forest problems can be solved disjointedly from overall problems is misleading. It is critical that everyone involved understands the need to cooperate for changing of country’s policies to protect forests and other natural resources concurrently with the work that will be implemented on the ground. 

Author Response

Thank you for these comments. As we noted in our revisions, we acknowledge that these structural problems are important. We do not feel there is room to go into the details, but provide references. We then place this analysis in this context by noting that it has been argued that addressing capacity issues may help to address these more structural issues.

We have further edited this part of the introduction, which now reads:

"The larger structural problems within the DRC are well-recognised [47] and political instability has resulted in poor, unstable governance and limited and contradictory policy generally and around forest management [19,41]. A full discussion of these issues is beyond the scope of this paper, but it is important to note that these issues directly limit effective forest management and need to be addressed if resource management is to be sustainable in the DRC. We do not suggest that these problems can be easily solved and we acknowledge that these structural problems exist, but we note that successful action is possible, despite these challenges [48]. In fact, it has been argued that the structures put in place through REDD+ or other payments for ecosystem services schemes may help countries address wider structural governance issues [49]. Here we discuss the role of capacity building in this context, to investigate its importance for REDD+ and more generally. Significant effort has been put into building structures and processes. Ensuring that those involved have the necessary capacity will be key to both REDD+ action in the current context [21] and the wider issue of addressing these structural challenges. This paper contributes to this by improving our understanding of what type of capacity building is needed at the provincial and community level for REDD+."

Note that we do not intend to decouple our analysis from these issues, but the  focus is determined by what the stakeholders in the workshops talked about. Although these wider structural issues did arise in some of the discussions, they were not the main focus of the discussions. Our analysis can only be guided by the data we collected.

In the conclusion, we link the capacity issues our analysis reveals back to the structural problems. Again, we have edited this part of the text to make this clearer:

"The REDD+ strategy is limited in how much it can address some of these issues that stem from political instability and lack of political will, and it recognises this. Greater awareness of, and capacity to address, these issues is needed to ensure the sustainable management of forests. This research shows that both the local stakeholders and the community are seeking knowledge and training to address these issues and capacity building is a key part of this. The next phase of national carbon market development needs to prioritise resolving systemic capacity building bottlenecks. Education and training in policy, management, governance, and sustainability more generally, needs to be an important part of REDD+. Such systemic capacity building calls for durable strategy to invest in higher education and vocational training and that recognise and understand the local context. Current and future REDD+ projects in the DRC risk failure because although they build structures and processes, if capacity building is ignored there are no people to populate those structures."

Addressing capacity and education issues will be a key part of addressing these structural problems. REDD+ currently fails to address capacity issues well, and we suggest this could be improved.

Reviewer 4 Report

The authors have revised the manuscript according to my previous comments. I suggest it can be accepted for publicatoin for this journal.

Author Response

We thank the reviewer for these positive comments.